# PUEGM: A Method of User Revenue Selection Based on a Publisher-User Evolutionary Game Model for Mobile Crowdsensing

**DOI:** 10.3390/s19132927

**Published:** 2019-07-02

**Authors:** Zihao Shao, Huiqiang Wang, Guangsheng Feng

**Affiliations:** College of Computer Science and Technology, Harbin Engineering University, Harbin 150001, China

**Keywords:** mobile crowdsensing, user revenue selection, publisher-user evolutionary game model (PUEGM), data quality assessment

## Abstract

Mobile crowdsensing (MCS) is a way to use social resources to solve high-precision environmental awareness problems in real time. Publishers hope to collect as much sensed data as possible at a relatively low cost, while users want to earn more revenue at a low cost. Low-quality data will reduce the efficiency of MCS and lead to a loss of revenue. However, existing work lacks research on the selection of user revenue under the premise of ensuring data quality. In this paper, we propose a Publisher-User Evolutionary Game Model (PUEGM) and a revenue selection method to solve the evolutionary stable equilibrium problem based on non-cooperative evolutionary game theory. Firstly, the choice of user revenue is modeled as a Publisher-User Evolutionary Game Model. Secondly, based on the error-elimination decision theory, we combine a data quality assessment algorithm in the PUEGM, which aims to remove low-quality data and improve the overall quality of user data. Finally, the optimal user revenue strategy under different conditions is obtained from the evolutionary stability strategy (ESS) solution and stability analysis. In order to verify the efficiency of the proposed solutions, extensive experiments using some real data sets are conducted. The experimental results demonstrate that our proposed method has high accuracy of data quality assessment and a reasonable selection of user revenue.

## 1. Introduction

With the development of intelligent terminal devices and wireless networks, MCS has become a frontier research issue for cross-space and large-scale data sensing [1]. MCS recruits a group of users to collect sensing data through their handheld devices, and the aggregation of the sensing data enables diverse services ranging from urban environmental monitoring and indoor map building to traffic information statistics and intelligent navigation [2,3]. The popularity of smartphones has accelerated the widespread use of MCS, but the presence of malicious and low-quality users will lead to a reduction in the effectiveness of the MCS platform.

Up to now, the research works on MCS user revenue mainly focused on the following aspects: revenue maximization [4,5,6,7], which aims at encouraging users to participate in the sensing activity, and data quality assessment [8,9,10,11,12,13,14], which determines how to select a suitable set of users to finish a sensing task. However, almost all the above works ignore two important factors: human bounded rationality and learning capacity. Simon pointed out that human bounded rationality refers to rationality between complete rationality and incomplete rationality under certain limits [15]. There are two reasons for people’s bounded rationality: Firstly, people usually face a complex and uncertain world. The more information exchanged, the greater the uncertainty of information, and this ultimately leads to the asymmetry of information. Secondly, people’s ability to calculate and recognize the environment is limited, and people cannot be omniscient. Learning capacity refers to the ability to observe and participate in new experiences, integrate new knowledge into existing knowledge, and thus change existing knowledge structures. That is to say, a user may improve the quality of data if he wants to achieve more revenue, even though the user may have different understanding and cognitive abilities. Meanwhile, existing researches on user revenue are rarely combined with their data quality assessment, which reduce the accuracy of user revenue strategy selection.

Revenue maximization and data quality assessment is not an easy thing in the MCS. For revenue maximization, incentive mechanisms to maximize the revenues of MCS are a hot issue (i.e., profit-based mechanisms [4,5] and reputation-based mechanisms [6,7]). In the research on incentive mechanisms, game theory is an important approach to maximize the revenues of MCS. Nowadays, most of game studies in MCS rely on the assumption of individual complete rationality, which is not consistent with the actual situation. Neglecting the limitations of bounded rationality, modeling and analysis of the behavioral game between the two sides will deviate from the actual situation, thereby reducing the accuracy and practical guiding value of selecting the optimal revenue strategy method. Evolutionary games have a dynamic ability to analyze the game behavior of both sides with bounded rationality, which takes the bounded rationality as the basis of game analysis. It can improve the inherent drive of the behavior strategy through learning mechanism, which can effectively enhance the accuracy and credibility of game model analysis.

Data quality assessment is also a challenging issue in MCS. Current studies on data quality are divided into incentive strategies [9,10,11] and classification strategies [8,12,13]. The research on incentive strategies is aims to encourage users to submit high-quality data by using appropriate incentives. Classification strategies focus on the data analysis stage, using data mining, machine learning and other methods to evaluate the data and filter out abnormal data to improve the data quality. Generally, users’ final revenue is positive proportional to their data quality [11]. If the quality of the user data is not evaluated, honest and high-quality users’ revenue will be affected, which may reduce the effectiveness of MCS. Therefore, the assessment of data quality can not only ensure users’ revenue, but also enhance the effectiveness of MCS.

Based on our previous research in data quality evaluation [16], we propose a user revenue selection for mobile crowdsensing which constructs the publisher-user game model by considering the bounded rationality and data quality, taking advantage of evolutionary game theory. The specific steps are as follows: Firstly, the choice of user revenue is modeled as a Publisher-User Evolutionary Game Model (PUEGM). Secondly, to remove low-quality data and improve overall quality of user data, we propose a data quality assessment algorithm in the PUEGM. Finally, the selected users are solved by the evolutionary strategy solution (ESS) and the stability analysis is carried out to obtain the optimal user revenue strategy under different conditions. Compared with [8], our research has the following advantages. First, we evaluate the quality of user data from view of reducing errors and abnormal data, which aims to detect low-quality and abnormal data with high accuracy. Meanwhile, we can distinguish data with different task requirements by weighting factors. Second, we construct the PUEGM from the perspective of human limited rationality. The game and evolution trends of the behaviors of both publishers and users are comprehensively analyzed and deduced, and the selection of user revenue strategies are analyzed and realized. Different from the existing work in [17], our research considers the effect of data quality on user revenue strategy and maximize the user revenue by improving the data quality. In addition, we derive the ESS of the MCS evolutionary game model and verify it with an example.

Our major contributions may be summarized as follows.

(1).Publisher-User Evolutionary Game Model (PUEGM): We modeled benefits of both publishers and users as evolutionary games, and deduced the evolutionary stability strategy (ESS), which can determine the choice of user revenue strategy.(2).Analyzing the impact of user data quality on its revenue in MCS, the ultimate user revenue is closely related to the data quality, and it is difficult to guarantee user reasonable revenue in the absence of data quality assessment. We analyzed the impact of user data quality on its revenue. The results can be applied to the user revenue selection under normal circumstances.(3).User data quality assessment algorithm. We propose an algorithm to remove the quality of low and abnormal data. At the same time, weighting factors are introduced to meet the different data quality concerns. We evaluated and compared the rationality of the proposed algorithm for classifying data quality and recognizing abnormal data.(4).Finally, as far as we know, this is the first time that the user revenue strategy is selected from the point of view of removing low and abnormal data quality in MCS data quality assessment. We think this is a tentative work for MCS research.

The rest of the paper is organized as follows: Section 2 discusses related work. Section 3 introduces the PUEGM and problem statement. Data quality assessment algorithm and user optimal revenue strategy selection are presented in Section 4. In Section 5, we evaluate our proposed method and present evaluation results. Finally, we conclude this paper in Section 6.

## 2. Related Work

Revenue maximization and data quality assessment are two important factors in MCS participants’ selection and hence there are several selection methods and models suggested in the literature. In this section, we discuss the state of the art in MCS participants’ selection and the various factors that are considered in those approaches.

### 2.1. Game Theory on MCS

Although the research on the revenue of MCS based on game theory has become a hotspot in recent years, few models are based on bounded rationality. Researchers have built a variety of MCS revenue models to address optimal revenues of both publishers and users. For example, Nie et al. [4] investigated an optimal incentive mechanism for a crowdsensing service provider, which is based on a Stackelberg game. It can effectively and efficiently recruit a sufficient number of mobile users to increase the revenue of public service providers. Pouryazdan et al. [6] believed that appropriate incentives for users can increase the value of data and propose an intelligent game method, which is to ensure the credibility of users from the perspective of data credibility and authenticity. To solve the interaction problem between the MCS server and multiple smartphone users, Xiao et al. [7] developed a Stackelberg game and used a deep Q-network to derive the best MCS strategy to prevent counterfeit sensor attacks. Similarly, game theory has been widely used in vehicle networks, Xiao et al. [5] deduced the Nash equilibrium between sensing accuracy and overall payment of MCS server in the static vehicle game, showing the trade-off between sensing accuracy and the overall payment of MCS server. 

As an important part of game theory, evolutionary gamed have a dynamic ability to analyze the game behavior of both sides with bounded rationality, which take the bounded rationality as the basis of game analysis. Evolutionary game can improve inherent drive of behavior strategy through learning mechanism, which can effectively enhance the accuracy and credibility of game model analysis. Although the existing researches on evolutionary game model most focus on network attack and defense behavior analysis [18], dynamic cloudlet selection and resource allocation [19] and achieve cluster stability in VANETs [20], evolutionary game model is rarely used in the field of MCS. It mainly includes an evolutionary game model proposed by [16] to describe the cooperative game phenomena in MCS networks and Ruan et al. [21] used an evolutionary game model to guarantee the risk of DoS attacks on MCS networks with minimum resource cost.

### 2.2. Quality-Oriented MCS

The quality issue of MCS has drawn many researchers’ attention in the past several years. In [14], the authors summarize the research on the quality of information in existing MCS, and analyze three important challenges (i.e., How can we trust that the smartphone-human sensors will send useful information? How can we enforce the submission of useful information? How can we estimate the usefulness of the submitted information?). The great differences among the qualities of the users’ data can be caused by both device factors and human factors in general. Device factors refer to different brands of mobile devices with different sensors and perceptions. Human factors refer to the complexity and unpredictability of human behavior. For example, some honest users will strictly obey the task instructions, while some careless or malicious users may use the wrong measurement methods to obtain some low-quality or wrong data.

Current studies on data quality are divided into incentive strategies [9,10,11] and classification strategies [8,12,13]. Incentive strategy refers to the use of appropriate incentives to encourage users to submit high-quality data. Song et al. [9] built an auction-based budget feasible mechanism to maximize the valuation of the performed tasks, which depends on the quality of sensing of users. Yang et al. [10] believe that the incentive value obtained by participants through providing data depends largely on the behavior of their social friends and proposed a social incentive mechanism. Peng et al. [11] improved the incentive mechanism from service quality and service providers’ interests, which not only guarantees the quality of data, but also can improve the overall profit. Classification strategy focuses on data analysis stage, using data mining, machine learning and other methods to evaluate data and filter out abnormal data to improve data quality. Liu et al. [12] proposed a context-aware data quality estimation method. This method confirms the relationship between context information and user data quality through historical data, and realizes the recruitment of users, which improves the overall efficiency of the mobile crowdsensing platform. In [8], the authors combine data quality classification and incentive mechanism methods to design an unsupervised learning method for data quality classification, which uses outlier detection to filter out abnormal data items. The incentive sharing mechanism is used to model the residual sharing process as a cooperative game, and the shapely value method is used to determine the payment for each user. Wang et al. [13] proposed a user selection utilizing data properties in mobile crowdsensing (SPM), where a triple-layer structure considering not only the temporal and spatial probability, but also the data’s property is formulated. This method can finish the largest number of sensing tasks.

While the existing approaches have their merits, they usually rely on specialized game models that only represent some of the aspects impacting the MCS participants’ selection process or are geared toward specific MCS applications. A game model that considers the dynamic and bounded rationality of the user selection strategy is important for MCS user revenue. In addition, the model should identify malicious users to protect the user’s reasonable revenue.

## 3. Publisher-User Evolutionary Game Model (PUEGM) and Problem Statement

In MCS, publishers and users have different decision-making mechanisms because of their various levels of cognition and skill, which lead to diverse benefits for participants. As time goes by, low-revenue users will constantly improve their strategies to achieve high-revenue, which is driven by the traction and learning mechanism based on the difference of revenue. 

### 3.1. Model Assumptions

We propose a method of user revenue selection based on the PUEGM for mobile crowdsensing. The objectives of the proposed method are to consider the impact of human limited rationality on revenue selection in mobile crowdsensing environments and to define a novel method for evaluation of users’ quality considering malicious and low-quality users, which is more reasonable than the existing models in terms of error rate and wide applicability. We assume the following assumptions about the participants:(1).The game players are divided into two groups, namely, publishers and users. Publishers can select different incentive mechanisms for the same task to be released according to the actual needs, and users can obtain different benefits according to the incentive mechanism of publishers (multi-selection of player strategies.).(2).User’s data quality selection strategy and publisher’s incentive strategy change dynamically over time (strategy dynamics)(3).Publishers’ revenue is directly proportional to the size of incentive mechanism and users’ revenue is directly proportional to the quality of data (revenue relevance).

### 3.2. Model Construction

In our work, we consider a scenario in which the collection of sounds is used as an example to collect information about the MCS data. Referring to the typical MCS structure, our MCS structure consists of three major components: task publishers (we will refer as “publishers” for simplicity), data providers (we will refer as “users” for simplicity), and a cloud platform. Publishers provide the required tasks and related incentive mechanisms to the cloud platform. Cloud platform will release relevant information to users. At the same time, users use their mobile devices to collect data and upload data to the cloud platform. The cloud platform will gain revenue for users who meet the task requirements through the quality of data. At last, the selected data are transmitted by the cloud platform to publishers, and both publishers and the selected users benefit from the data.

*Step 1*. Define the Publisher-User evolutionary game model

In our work, we define the Publisher-User evolutionary game model as a four-tuple. 

**Definition** **1.**
*Define the PUEGM as a four-tuple, i.e., PUEGM = (N, W, R, B).*
*(1)*.
*N = (N_u_, N_p_) is the participants of both evolutionary games, where N_u_ denotes the user, and N_p_ denotes the publisher.*
*(2)*.
*W = (uw, pw) is the strategy space of both sides, where uw = {uw_1_, uw_2_, …, uw_n_} denotes users’ data quality optional strategy sets, and pw = {pw_1_, pw_2_, …, pw_n_} denotes publishers’ task-motivated optional strategy sets.*
*(3)*.
*R = (d, q) is the game belief set, where d_i_ denotes the probability that publishers choose pw_i_, and q_j_ denotes the probability that users select uw_j_.*
*(4)*.
*B = (B_u_, B_p_) is the set of the revenue functions of both sides, where B_u_ denotes the revenue of the user and B_p_ denotes the revenue of the publisher.*



### 3.3. Evaluate the Quality of User Data

In [11], the authors prove that the user’s final revenue is positive proportional to user overall data quality. Therefore, we use the error-eliminating decision-making method [22] to improve the quality of user data in the PUEGM, and add assessment of data quality in this model, which ensures that high-quality user data can be better applied and maximizes the value of $\bar{B_{u}}$. When users receive sensed tasks from the platform, they will use the mobile device to collect data. The collected data contains a lot of useful information because of the variety of sensors embedded in the mobile device, such as location, time and etc. In the data quality assessment method, we have data for *m* users, denoted by $U=\{u_{1},u_{2},\ldots ,u_{m}\}$. We quantify each user data quality strategy sets into multiple data quality assessment indexes, denoted by $C=\{c_{1},c_{2},\ldots ,c_{m}\}$, and the decision matrix is denoted by $G={\left[g_{i,j}\right]}_{n\times m}$, where $g_{i,j}$ represents numerical value of $u_{i}$ under $c_{j}$.

The quality may be directly linked to the timeliness and accuracy of sensor measurements. In addition, high-quality collection environment and reliable collection methods can be given priority in order to guarantee the reliability of data.

Hence, the values of different sensors may be calculated using so-called quality valuation functions which are used to rank sensors from “best” to “worst”. More formally, we consider the overall value of sensors be *R_i_*, expressed as a weighted combination of multiple value dimensions, namely response time ($c_{1}$), distance ($c_{2}$), error-free data integrity ($c_{3}$) and data reliability ($c_{4}$) [23]. Different weights $\omega_{j}$ indicate the relative importance of a given value dimension in contributing to overall sensor value, and may be considered to be application specific.

In our case, to quantify each of the value dimensions, there is a need to consider the underlying factors influencing each of these dimensions. In practice, sensor metadata delivered to the platform will provide necessary input parameters (e.g., location, time, technical characteristics) for the valuation function, as discussed in the following:

Response time ($c_{1}$): is a metric indicating the time difference between the time of publishing and submitting task.

Distance ($c_{2}$): is a metric indicating the distance of users from the geographic location with respect to the specified MCS task. For example, if a given monitoring application requires sensor readings at a certain location, then it will be necessary to consider the distance of the sensor reading with respect to the target location.

Data integrity ($c_{3}$): is a quality measure that can be defined as the accuracy of data transmitted by users in different states. For example, in the application of noise monitoring, the noise monitoring results transmitted by users in static state will be much better than the data transmitted in running state.

Data reliability ($c_{4}$): is a quality measure that can be defined as the accuracy of data transmitted by users in different environments. For example, in the application of noise monitoring, users are more willing to monitor quiet environments, while in noisy environments they are more likely to provide low-quality data to reduce the pain of noise.

The assessment indexes of data quality are divided into negative correlation and positive correlation [24]. The negative correlation indexes refer to the higher data quality of users, the smaller value of the assessment indexes. The positive correlation indexes refer to the lower data quality of users, the smaller value of the assessment indexes. In this paper, we record the data quality assessment indexes as the cost index and benefit index, denoted by $C_{cost}$ and $C_{profit}$, which satisfies $C_{cost},C_{profit}\subseteq C$, $C_{cost}\cup C_{profit}=C$, $C_{cost}\cap C_{profit}=\emptyset$. When the expected range of each data quality assessment indexes are determined, the error elimination rule is used to judge user data and the error value is used to evaluate user data quality. The detailed steps are as follows:

Firstly, we determine the user data quality assessment indexes and calculate the error value of user data by equations (1) and (2):(1)$$t_{i,j}=\{\begin{array}{c} \rho ,g_{i,j}>z_{j}^{\max}org_{i,j}<z_{j}^{\min}\\ \frac{g_{i,j}-z_{j}^{\min}}{z_{j}^{\max}-z_{j}^{\min}},z_{j}^{\min}\leq g_{i,j}\leq z_{j}^{\max} \end{array},$$
(2)$$t_{i,j}=\{\begin{array}{c} \rho ,g_{i,j}<z_{j}^{\min}\begin{array}{c} org_{i,j}>z_{j}^{\max} \end{array}\\ \frac{z_{j}^{\max}-g_{i,j}}{z_{j}^{\max}-z_{j}^{\min}},z_{j}^{\min}\leq g_{i,j}\leq z_{j}^{\max} \end{array},$$
where i=1,2,…,n,j=1,2,…,m, and $t_{i,j}$ represents the error value of $u_{i}$ under $c_{j}$. The expected lower and upper limit of data quality assessment indexes are expressed as $z_{j}^{\min}$ and $z_{j}^{\max}$. When $c_{j}\in C_{cost}$, we use Equation (1) to calculate the error value of user data. When $c_{j}\in C_{profit}$, we use Equation (2) to calculate the error value of user data. The error value is represented by a constant ρ. In order to distinguish the value of ρ and ensure the same order of magnitude, we specify ρ=1+ε and ε is a positive infinite decimal. Through equations (1) and (2), the sequence of error value of the user data $u_{i}$ under different data quality assessment indexes can be obtained, which is recorded as $\left\{t_{i,1},t_{i,2},\ldots ,t_{i,m}\right\}$.

Secondly, we calculate the maximum error value ($t_{i}^{\max}$) and judge rationality, which can be expressed as:(3)$$t_{i}^{\max}=\max\{t_{i,j}\}.$$

When $t_{i}^{\max}=\rho$, $u_{i}$ is the erroroneous data and this data is removed.

Thirdly, we standardize data quality assessment indexes and calculate the limit error value of data quality assessment indexes ($E_{j}^{*}$), which can be expressed as:(4)$$\{\begin{array}{c} Z_{j}^{\max}=\frac{z_{j}^{\max}}{z_{j}^{\max}}=1,Z_{j}^{\min}=\frac{z_{j}^{\min}}{z_{j}^{\max}}\\ E_{j}^{*}=\frac{{\left(Z_{j}^{\max}-Z_{j}^{\min}\right)}^{2}}{\sum_{i=1}^{n}{\left(Z_{i}^{\max}-Z_{i}^{\min}\right)}^{2}} \end{array}.$$

Finally, we calculate and sort the error loss sequence ($r_{u_{i}}$) to achieve the optimal user choice.

Different tasks often need various data. For example, in the fastest route search traffic monitoring of rescue vehicles, the required data pays more attention to the accuracy and effectiveness, which means that the weight of location and time parameters in data quality indexes is larger. In the environmental pollution detection, more attention will be paid to the integrity and accuracy of the data, which means that the weight of integrity and transmission accuracy parameters in data quality indexes is larger. Therefore, weighting $\omega_{j}$ is introduced to achieve the change of attention to different task requirements. The error loss sequence of user data is as follows:(5)$$r_{u_{i}}=\{\omega_{1}\cdot t_{i,1}\cdot E_{1}^{*},\omega_{2}\cdot t_{i,2}\cdot E_{2}^{*},\ldots ,\omega_{j}\cdot t_{i,j}\cdot E_{j}^{*}\},$$
where $i=1,2,\ldots ,n,j=1,2,\ldots ,m,\sum_{j=1}^{m}\omega_{m}=1$.

The error loss sequence $r_{u_{i}}$ will be seen as the n-dimensional space on the point. Higher data quality means the value of *R_i_* is closer to the origin and the value of *R_i_* is arranged in ascending order. Similarly, we can set the threshold of *R_i_* to remove low quality user data. The equation is:(6)$$R_{i}=\sqrt{\sum_{j=1}^{n}{\left(\omega_{j}\cdot t_{i,j}\cdot E_{j}^{*}\right)}^{2}},i=1,2,\ldots ,n.$$

### 3.4. Analyze the User Optimal Revenue Strategy

Through the previous section, we can not only remove malicious and low-quality user data, but also achieve a reasonable user data quality strategy set. We combine the publisher’s task to stimulate the optional strategy set, solve the evolution strategy solution and conduct the stability analysis, and finally obtain the user optimal revenue strategy under different conditions. 

In PUEGM, *N_u_* and *N_p_* have multiple choices. The probability that the same strategy adopted by both players may also be different, which in different stages of the game. The decision makers of both sides will constantly adjust their strategies under the influence of revenues, so that the number of decision makers who choose different strategies will change, and the process of the game between the two sides will be characterized by dynamic evolution. The publisher-user game tree is shown in Figure 1.

When different strategies are adopted, both players will generate different revenue values. The payoff matrix is expressed as follows:(7)$$\left\{\begin{array}{cccc} a_{11},b_{11} & a_{12},b_{12} & \cdots & a_{1n},b_{1n}\\ a_{21},b_{21} & a_{22},b_{22} & \cdots & a_{2n},b_{2n}\\ \vdots & \vdots & \ddots & \vdots \\ a_{m1},b_{m1} & a_{m2},b_{m2} & \cdots & a_{mn},b_{mn} \end{array}\right\},$$
where $a_{ij}$ represents the publisher’s revenue when publisher adopts *i* strategy and user adopts *j* strategy, $b_{ij}$ represents the user’s revenue when user adopts the *j* strategy and publisher adopts *i* strategy, $q_{1}+q_{2}+\cdots +q_{n}=1$ and $d_{1}+d_{2}+\cdots +d_{m}=1$.

Based on the above conditions, we calculate the expected (i.e., $B_{uw_{n}},B_{pw_{m}}$) and average payoffs (i.e., $\bar{B_{u}},\bar{B_{p}}$) of the user and the publisher:(8)$$B_{uw_{n}}=d_{1}b_{1n}+d_{2}b_{2n}+\cdots +d_{m}b_{mn}=\sum_{j=1}^{m}d_{j}b_{jn}$$
(9)$$B_{pw_{m}}=q_{1}a_{m1}+q_{2}a_{m2}+\cdots +q_{n}a_{mn}=\sum_{i=1}^{n}q_{i}a_{mi}$$
(10)$$\bar{B_{u}}=q_{1}B_{uw_{1}}+q_{2}B_{uw_{2}}+\cdots +q_{n}B_{uw_{n}}=\sum_{i=1}^{n}q_{i}B_{uw_{i}}$$
(11)$$\bar{B_{p}}=d_{1}B_{pw_{1}}+d_{2}B_{pw_{2}}+\cdots +d_{m}B_{pw_{m}}=\sum_{i=1}^{m}d_{i}B_{pw_{i}}$$

In PUEGM, to achieve more revenues, low-income users will improve their data quality and change their strategies, which lead to a change in the proportion of user who choose different data quality strategies over time. We use $q_{i}(t)$ to indicate the proportion of user who choose the strategy $uw_{i}$, where $\sum_{i=1}^{n}q_{i}(t)=1$. For a certain data quality optional strategy $uw_{i}$, the dynamic rate of change can be denoted by replicator dynamic equation [25], which is written as:(12)$$u(q)=\frac{dq_{i}(t)}{dt}=q*(B_{uw_{i}}-\bar{B_{u}}).$$

At the same time, to maximize the benefits of publishers, they may change incentive strategies according to the quality of data provided by users, which leads to the proportion of selecting different strategies changes dynamically over time. We use $d_{i}(t)$ to indicate the proportion of publisher who choose the strategy $pw_{i}$, where $\sum_{i=1}^{n}d_{i}(t)=1$. For any publishers’ strategy, the corresponding replicator dynamic equation is defined as:(13)$$p(d)=\frac{dd_{i}(t)}{dt}=d*(U_{pw_{i}}-\bar{U_{p}}).$$

Let $F=\left[\begin{array}{c} u(q)\\ p(d) \end{array}\right]=0$, we can get duplicate dynamic equations and use the Jacobian matrix local stabilization method to perform stability analysis on all equilibrium points. Finally, the stable equilibrium solution of this model is obtained, and the optimal user data quality strategy selection is analyzed.

### 3.5. Problem Formulation

User’s goal is to achieve greater benefits with a certain amount of data quality. In PUEGM, those *n* users can adopt different strategy to get the maximize revenue after the publisher publishes some sensing task. The strategy adopted by the publisher is known, i.e., the expected payoffs of the user in Equation (8) is known. We can achieve the user’s optimal strategy by increasing the average payoffs of the selected users, that is, maximizing $\bar{B_{u}}$. In Equation (9), the value of $\bar{B_{u}}$ is related to *q_i_* and *q_i_* denotes the probability that users select *uw_i_*. User’s final revenue is positive proportional to user overall data quality. To maximize the user revenue, we translate the problem of solving $\bar{B_{u}}$ into an assessment of user data quality. We formalize it as:(14)$$\bar{B_{u}}=f(\omega_{1}\cdot t_{i,1}\cdot E_{1}^{*},\omega_{2}\cdot t_{i,2}\cdot E_{2}^{*},\ldots ,\omega_{j}\cdot t_{i,j}\cdot E_{j}^{*})$$
where $\omega_{j}$ is the weighting factor, the value of $t_{i,j}$ and $E_{j}^{*}$ can be calculated by equations (1), (2) and (4). In our user data quality assessment, the value of this function is related to *R_i_*, for which we define it as:(15)$$f(\omega_{1}\cdot t_{i,1}\cdot E_{1}^{*},\omega_{2}\cdot t_{i,2}\cdot E_{2}^{*},\ldots ,\omega_{j}\cdot t_{i,j}\cdot E_{j}^{*})=R_{i}=\sqrt{\sum_{j=1}^{n}{\left(\omega_{j}\cdot t_{i,j}\cdot E_{j}^{*}\right)}^{2}},i=1,2,\ldots ,n$$

Therefore, our main purpose is seek an evolution game model which can choose the user’s optimal revenue strategy. In the case of the both strategies are given, the evolution game model can quickly evolve into a robust and stable system in the case of any given strategy probability and obtain the optimized user revenue. What’s more, the proposed data quality assessment method based on the evolution game model can answer the following question. For publishers, it can predict in advance how many are the participant in the game? How about the data quality of each user?

## 4. Algorithm Introduction and User Revenue Analysis

### 4.1. Data Quality Assessment Algorithm

A simple and effective algorithm, as shown in Algorithm 1, is used to evaluate the quality of users’ data, which is based on an index that quantifies the quality of existing data. 

**Algorithm 1:** Implementation of user data quality assessment**Input**: User data**Output**: Sorted the user data quality1: **Initialization**2: Determine the user data quality assessment indexes *c_j_*3: **for**
*i* = 1,2,… ,*n*, *j* = 1,2,…,*m* do4:  Calculate *g_i,j_*5:  Judge if *c_j_* belongs to the cost index or benefit index6:  **if**
*c_j_* ∈ *C_cost_*7:   Calculate the error value of data by Equation (1)8:   **else**
9:   Calculate the error value of data by Equation (2)10:  **end if**11:    Calculate $t_{i}^{\max}$ by Equation (3)12:  **if**
$t_{i}^{\max}=\rho$
13:   Calculate $E_{j}^{*}$ and $r_{u_{i}}$ by equations (4) and (5)14:   Sort the user data using Equation (6)15:  **else**16:   Remove errors and low-quality data17: **end for**

When users receive the task, they use smartphone to collect relevant data to generate a user data set, denoted by *U* = {*u_1_*, *u_2_*, …, *u_m_*}. In the assessment of user data quality, we remove low-quality data and sort reasonable user data by descending the loss of errors. At the same time, we also introduce weighting factors to enhance the usability of the algorithm.

The complexity of the algorithm is O(mn). From 1 to 2, the algorithm is used to determine the type of users’ data quality assessment indexes, which aims to quantify each user’s data quality strategy sets. From 3 to 13, it is used to judge whether users’ data will be correct in different quality assessment indexes and obtain errors value of data, which is based on the error elimination decision-making method. From 13 to 17, it is used to sort user data, which can remove low-quality data and select high-quality data. 

Algorithm1 realizes the evaluation of user data quality and removes low-quality data. This method provides a guarantee for user revenue by improving the proportion of high-quality data. We believe that the data quality assessment indexes of MCS can be improved continuously according to the need in future research, so that the assessment of data quality of MCS is more accurate. Therefore, the data quality assessment method proposed in this paper has good scalability by the change of assessment indexes.

### 4.2. User Revenue Optimal Strategy Selection Algorithm

In user revenue optimal strategy selection algorithm, first, a publisher-user evolutionary game tree is constructed to determine the payoff matrix of game, which is denoted by *uw* and *pw*. Then, we achieve the ESS solution by computing $B_{uw_{n}}$, $B_{pw_{m}}$, $\bar{B_{u}}$ and $\bar{B_{p}}$. Next, we construct the replication dynamic equation and the Jacobian matrix. Finally, we obtain the optimal revenue strategy based on the ESS. ESS is a strategy such that, if most of the members of a population adopt it, there is no “mutant” strategy that would give higher reproductive fitness [26]. An ESS is an equilibrium refinement of the Nash equilibrium. It is a Nash equilibrium that is “evolutionarily” stable: once it is fixed in a population, natural selection alone is sufficient to prevent alternative (mutant) strategies from invading successfully. This means that when the user is in the ESS state, the user’s optimal revenue can be achieved. 

In our case, the basic idea of obtaining the optimal user revenue strategy is threefold. Firstly, the publisher-user game tree is constructed to show that publishers and users have their own optional strategy sets and the probability of strategy selection is different; secondly, the corresponding replication dynamic equation is established based on the evolutionary game theory; finally, the optimal user revenue strategy is obtained by calculating the evolutionary stable equilibrium solution. The specific algorithm is as follows.

**Algorithm 2:** Implementation of user optimal revenue strategy**Input**: Publisher-user game tree**Output**: Users’ optimal revenue strategy1: **Initialization**2: Build users’ type space collection $U=\{u_{i},i\geq 1\}$ and optional strategy space collection $uw=\{uw_{j},1\leq j\leq m\}$
3: Select reasonable users’ strategy $uw_{i}$ with probability $q_{i}(1\leq i\leq n)$, where $\sum_{i=1}^{n}q_{i}=1$4: Calculate $b_{ij}$
5: Calculate $B_{uw_{i}}$ and $\bar{B_{u}}$ by Eqs. 8 and 106: Establish users’ replication dynamic equation *u*(*q*) and calculate the evolution equilibrium point7: Construct a Jacobian matrix to analyze the stability of the equilibrium point and obtain a stable equilibrium solution8: Output users’ revenue strategy9: **End**

#### 4.2.1. Analysis of Algorithm Time Complexity

From 1 to 5, the time complexity is O(m+n). From 6 to 9, the time complexity is O((m+n)^2^). Therefore, the time complexity for solving evolutionary stable equilibrium solution is O((m+n)^2^).

#### 4.2.2. Analysis of Spatial Complexity

The storage space consumption is mainly concentrated on the storage of the intermediate values of strategy benefit and equilibrium solution, so the space complexity is O(mn).

Algorithm 2 obtain the revenue of each strategy, the evolution steady state by analyzing game results, and realize prediction and assessment of strategy selection mechanism. This is a mode of users’ strategy selection from the perspective of maximizing user revenue, which guides user decision-making. Since the number of policies can be extended, the method is versatile and can be applied to general users’ revenue strategy selection. In the next section, we will use a two-category strategy for verification.

### 4.3. Example Description

We assume that publishers are *p*, users are *u*. $\left\{pw_{l},pw_{h}\right\}$ and $\left\{uw_{l},uw_{h}\right\}$ is a simple strategy adopted by both sides. Publishers’ incentive strategy and users’ data quality are divided into two categories, namely low and high. The probabilities of selection of different strategies are different and generate different revenues. When publishers adopt high incentive strategy and users adopt high data quality, the revenue of both players can be maximized. The definition of player revenue is as follows.

**Definition** **2.**
*The strategic revenue of both players should meet the following requirements, where $a_{hh}>a_{lh}>a_{ll}>a_{hl}$ and $b_{hh}>b_{hl}>b_{ll}>b_{lh}$.*


We calculate the expected and average payoffs of users, which using low and high strategies, denoted as:(16)$$\left\{ \begin{array}{l} B_{uw_{l}}=db_{ll}+(1-d)b_{hl}\\ B_{uw_{h}}=db_{lh}+(1-d)b_{hh}\\ \bar{B_{u}}=qB_{uw_{l}}+(1-q)B_{uw_{h}}=q[db_{ll}+(1-d)b_{hl}]+(1-q)[db_{lh}+(1-d)b_{hh}] \end{array}\right..$$

For user strategy, the replication dynamic equation is obtained according to Equation (12), denoted as:(17)$$u(q)=\frac{dq(t)}{dt}=q*(B_{uw_{l}}-\bar{B_{u}})=q(1-q)[d(b_{ll}-b_{hl}-b_{lh}+b_{hh})+(b_{hl}-b_{hh})].$$

When u(q)=0, we can get the equilibrium solution, i.e., q=0, q=1 and $d=\frac{b_{hh}-b_{hl}}{b_{ll}-b_{hl}-b_{lh}+b_{hh}}$.

**Proposition** **1.**
d∈(0,1)


**Proof.** By Definition 2, we can get $\{\begin{array}{c} b_{hh}-b_{hl}>0\\ b_{ll}-b_{lh}>0 \end{array}$, i.e., $b_{hh}-b_{hl}+b_{ll}-b_{lh}>b_{hh}-b_{hl}>0$. Combining $d=\frac{b_{hh}-b_{hl}}{b_{ll}-b_{hl}-b_{lh}+b_{hh}}$ and $b_{hh}-b_{hl}+b_{ll}-b_{lh}>b_{hh}-b_{hl}>0$, we have d∈(0,1). □

Similarly, we calculate the expected payoffs, average payoffs, and dynamic equations for publishers, which can be expressed as:(18)$$\left\{ \begin{array}{l} B_{pw_{l}}=qa_{ll}+(1-q)a_{lh}\\ B_{pw_{h}}=qa_{hl}+(1-q)a_{hh}\\ \bar{B_{p}}=dB_{pw_{l}}+(1-d)B_{pw_{h}}=d[qa_{ll}+(1-q)a_{lh}]+(1-d)[qa_{hl}+(1-q)a_{hh}] \end{array}\right.,$$
(19)$$p(d)=\frac{dd(t)}{dt}=d*(B_{pw_{l}}-\bar{B_{p}})=d(1-d)[q(a_{ll}-a_{hl}-a_{lh}+a_{hh})+(a_{lh}-a_{hh})].$$

When p(q)=0, we can get the equilibrium solution, i.e., d=0, d=1 and $q=\frac{a_{hh}-a_{lh}}{a_{ll}-a_{hl}-a_{lh}+a_{hh}}$.

**Proposition** **2.**q∈(0,1).

**Proof.** The method of proof is consistent with Proposition 1. ☐

Users and publishers’ dynamic equations are combined to construct publisher-user game evolution equations. When F=0, we can get five sets of equilibrium points, i.e., $F_{1}(0,0)$, $F_{2}(0,1)$, $F_{3}(1,0)$, $F_{4}(\frac{a_{hh}-a_{lh}}{a_{ll}-a_{hl}-a_{lh}+a_{hh}},\frac{b_{hh}-b_{hl}}{b_{ll}-b_{hl}-b_{lh}+b_{hh}})$ and $F_{5}(1,1)$.

We use the system dynamics method [27] to analyze the PUEGM. Meanwhile, we use the Jacobian matrix local stability method [28] to analyze the stability of all evolution equilibrium points and solve the stable equilibrium solution. According to the Jacobi matrix local analysis method, if detJ>0 and traceJ>0, the equilibrium point is unstable point. If detJ>0 and traceJ<0, the equilibrium point is stable point. If detJ<0, the equilibrium point is saddle point. Values of the Jacobi determinant and trace are shown in Table 1.

We obtain that *F*_1_ and *F*_5_ are stable points and *F*_2_ and *F*_3_ are unstable points. Since the value of equilibrium *F*_4_ is uncertain, we use the replication dynamic evolution graph to analyze it, as shown in Figure 2. According to Equation (18) and Figure 2, *F*_4_ has three stable states at most: 0, 1, and $\frac{b_{hh}-b_{hl}}{b_{ll}-b_{hl}-b_{lh}+b_{hh}}$, respectively. For any user’s strategy, when the value of *d* is 0 and 1, the tangent slope is negative and we can define it as a stable evolution strategy. When the value is $\frac{b_{hh}-b_{hl}}{b_{ll}-b_{hl}-b_{lh}+b_{hh}}$ and the probability *q* is chosen to satisfy $\frac{dq(t)}{dt}=0$, the state is stable too. When $d<\frac{b_{hh}-b_{hl}}{b_{ll}-b_{hl}-b_{lh}+b_{hh}}$, q=0 is the ESS of users. When $d>\frac{b_{hh}-b_{hl}}{b_{ll}-b_{hl}-b_{lh}+b_{hh}}$, q=1 is the ESS of users. Similarly, when $q<\frac{a_{hh}-a_{lh}}{a_{ll}-a_{hl}-a_{lh}+a_{hh}}$, d=0 is the ESS of publishers. When $q>\frac{a_{hh}-a_{lh}}{a_{ll}-a_{hl}-a_{lh}+a_{hh}}$, d=1 is the ESS of publishers. Therefore, this model has an evolutionary stable equilibrium solution.

## 5. Performance Evaluation

### 5.1. Basic Simulation Setup

In our experiments, the data we used came from the real Dartmouth College Wi-Fi campus trace data set [29], which was an experiment on the open source middleware NSense. This data takes sound collection as an example, including timestamps, distance between test points and sensing nodes, data collection methods (i.e., STATIONARY, WALKING, and RUNNING), and data collection environments (i.e., QUIET, NORMAL, ALERT, and NOISY). Data quality assessment indexes were divided into response time ($c_{1}$), distance ($c_{2}$), data integrity ($c_{3}$) and data reliability ($c_{4}$), which is based on the classification method of [23]. 

To ensure the diversity of data and improve the accuracy of the assessment, we quantify existing data and determine reasonable range as shown in Table 2. Our experiments are all implemented on MATLAB2013b.

### 5.2. Experiment Results of Data Quality Assessment

We use Algorithm 1 to realize the accurate identification and sorting of user data. Our previous work has proved that this algorithm can solve three problems: removing low quality data, selecting different demand data, and sorting data quality [17]. In this paper, we will supplement the experiment to show that our method can not only solve these three problems, but also has the advantages of abnormal recognition precision and high accuracy of data classification.

In MCS, high-accuracy data quality assessment means that the overall quality of data can be improved, so that users can achieve greater revenues. Our experiments collected more than 5000 data sets, and some abnormal data were added in the data set, which aims to simulate the error data in the actual application. We will compare the performance with two common anomaly detection methods (i.e., BP method and SVM method [30]), by calculating the abnormal data recognition precision (*X_da_*) and data classification accuracy (*X_dp_*), which can be expressed as:
(20)$$\left\{ \begin{array}{c} X_{da}=\frac{{sum}_{jad}}{{sum}_{ad}}\\ X_{dp}=\frac{{sum}_{jd}}{{sum}_{d}} \end{array},\right.$$
where *sum_jad_* represents the number of true abnormal data that are judged as abnormal data, *sum_ad_* represents the number of abnormal data, *sum_jd_* represents the number of data that are judged correctly, *sum_d_* represents the number of data. Comparison of abnormal data recognition precision and data classification accuracy is provided in Figure 3.

In the abnormal data recognition precision experiment, the augmentation of abnormal data increases the precision of BP method and SVM method. The reason is that both methods require a large number of training samples to achieve more accurate recognition. Meanwhile, the precision of our method is better than these two methods, and the accuracy rate is always 1. The purpose of our method is to remove false anomaly data, which can be very sensitive when the values deviate. This shows that our method has the advantage of high precision in recognizing abnormal data, small demand for training samples and only related to the boundaries of each type of data quality. 

In the data classification accuracy experiment, with the increase of the proportion of abnormal data, the accuracy of our method and SVM method in recognizing data is basically above 95%, which shows that our method has the advantages of high accuracy and anti-interference in data classification.

### 5.3. Experiment Results of User Revenue Optimal Strategy

We use system dynamics to simulate and validate the equilibrium solution of the PUEGM, then, we obtain the optimal revenue strategy of users in the PUEGM. Through the analysis of Section 4.3, we know that stable equilibrium solutions of the model are *F_1_*, *F_4_*, *F_5_*. In the following experiments, we change the initial state of *d* and *q* to obtain the evolutionary trend and the steady state of the evolutionary game, and finally achieve the optimal revenue strategy of users.

We assume that the benefits of publishers and users using different strategies in MCS are known, so the stable equilibrium solution ($F_{4}$) of this system can be confirmed (i.e., $F_{4}=\left[\begin{array}{c} 0.4\\ 0.6 \end{array}\right]$). At the same time, the incentive strategies provided by publishers are divided into low incentive strategy (*pw_l_*) and high incentive strategy (*pw_h_*). The data quality strategies provided by users will be divided into low data quality strategy (*uw_l_*) and high data quality strategy (*uw_h_*).

(1)When *d* = 0 and *q* = 0, it means that publishers all provide strategy *pw_h_*, and users all provide strategy *uw_h_*. According to the simulation, we find that strategies selection has no change with the evolution time, as shown in Figure 4. The evolution result can be one of the stable states of the system, which also verifies that the user revenue can be maximized when publishers adopt a high incentive strategy and users adopt a high data quality strategy.(2)When *d* = 0.6 and *q* = 0.4, it means that publishers choose strategies *pw_l_* and *pw_h_* with a probability of (0.6,0.4) and users choose strategies *uw_l_* and *uw_h_* with a probability of (0.4,0.6). According to the simulation, we find that strategies selection has no change with the evolution time, as shown in Figure 5. The evolution result can be one of the stable states of the system, which verifies that this state is the equilibrium point.(3)When *d* = 1 and *q* = 1, it means that publishers all provide strategy *pw_l_*, and users all provide strategy *uw_l_*, as shown in Figure 6. According to the simulation, we find that strategies selection has no change with the evolution time, because publishers only provide a low incentive strategy, and users can’t get higher revenues even if they provide high quality data. Therefore, users only choose strategy *uw_l_* to ensure the optimal revenue of users.(4)When *d* = 0.4 and *q* = 0.3, it means that publishers choose strategies *pw_l_* and *pw_h_* with a probability of (0.4,0.6) and users choose strategies *uw_l_* and *uw_h_* with a probability of (0.3,0.7), as shown in Figure 7. After multiple games between the two players, we find that both sides of the game tend to *F*_1_ and users can choose strategy *uw_h_* to obtain the optimal revenue, because publishers have a higher probability to adopt a high incentive strategy and user is more likely to provide high quality data to maximize their revenues.(5)When *d* = 0.7 and *q* = 0.6, it means that publishers choose strategies *pw_l_* and *pw_h_* with a probability of (0.7,0.3) and users choose strategies *uw_l_* and *uw_h_* with a probability of (0.6,0.4), as shown in Figure 8. After multiple games between the two players, we find that the stable equilibrium solution of both sides of this game tends to *F*_5_ and users can choose strategy *uw_l_* to obtain the optimal revenue, because publishers have a higher probability to adopt a low incentive strategy and users are more likely to provide low quality data to avoid the loss of their revenues.(6)When *d* = 0.4 and *q* = 0.6, it means that publishers choose strategies *pw_l_* and *pw_h_* with a probability of (0.4,0.6) and users choose strategies *uw_l_* and *uw_h_* with a probability of (0.6,0.4), as shown in Figure 9. Although, the stable equilibrium solution of both sides of the game tends to *F_4_* at the beginning, it tends to *F_5_* eventually. The reason for this result is that the values of *d* and *q* are very close to *F_4_* at the beginning, which leads to a balance between two players in a short time. However, in *F_4_*, the values of *d* and *q* can only be the initial equilibrium state, and users are not willing to provide high quality data. Therefore, users will eventually choose strategy *uw_l_* to achieve the best revenue.(7)We find that the stable equilibrium solutions of the two players are different in various initial states, as shown in Figure 10. When the value is 1, it means that both sides of the game will tend to *F_5_* and users can get the best revenue by choosing strategy *uw_l_*. When the value is 0, it means that both sides of the game will tend to *F_1_* and users can get the best revenue by choosing strategy *uw_h_*. When the value is 0.5, it means that the game will tend to *F_4_* and users select strategies *uw_l_* and *uw_h_* with a probability of (0.4,0.6) to achieve the best revenue. We find that the higher quality of user data, the more stable equilibrium solution tends to *F_1_*, when publishers strategy is unchanged.

It can be seen from the above simulation results that the initial state selected by different strategies will eventually reach a certain stable state through the evolution. Meanwhile, experimental results are consistent with the theoretical analysis in the model, which shows that the evolutionary game model is consistent with the development in reality world. Therefore, in MCS, the PUEGM can be used to analyze the selection of publishers’ incentive strategy and provide a basis for the selection of user optimal revenue strategy.

## 6. Conclusions

In this paper, the problem of selecting an optimal user revenue selection in mobile crowdsensing and data quality assessment has been investigated. On the one hand, the data quality assessment algorithm can solve three problems, including removing low quality data, selection of data for different needs and sorting data quality. Compared with the BP and SVM methods, this method has high accuracy and a wide application range. On the other hand, to motivate users to provide high quality data and prevent low quality data, we model the user revenue as an evolutionary game and determine the optimal revenue for each user by solving the evolutionary stability strategy solution. According to the needs of different scenarios, different data quality evaluation indexes can be established and the weight of indexes can be modified. Thus, we believe that the proposed model has a good scalability and practical significance. As for future work, reasonable selection of user data quality assessment indexes, user historical credibility and reasonable allocation of different index weights will be our main research content.

## Figures and Tables

**Figure 1 sensors-19-02927-f001:**
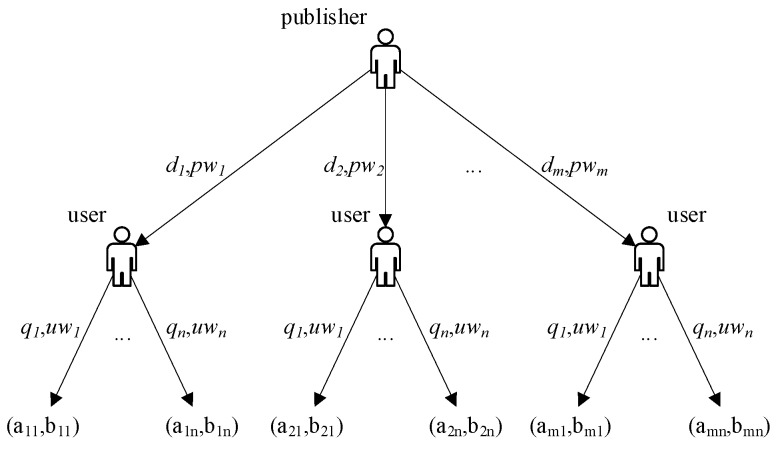
Publisher-user game tree.

**Figure 2 sensors-19-02927-f002:**
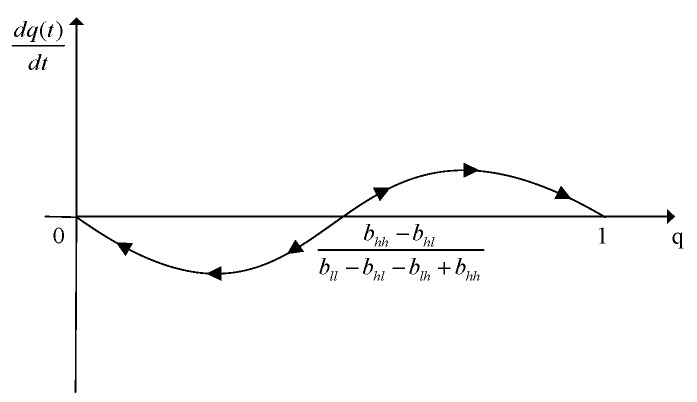
Replication dynamic evolution graph of user strategy.

**Figure 3 sensors-19-02927-f003:**
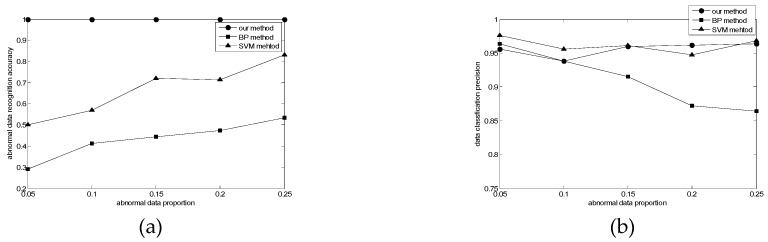
(**a**) Comparison of abnormal data recognition precision; (**b**) Comparison of data classification accuracy.

**Figure 4 sensors-19-02927-f004:**
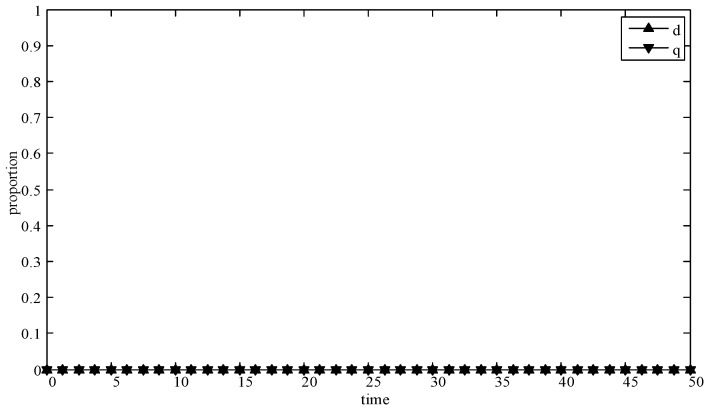
Evolution curves of *d* = 0 and *q* = 0.

**Figure 5 sensors-19-02927-f005:**
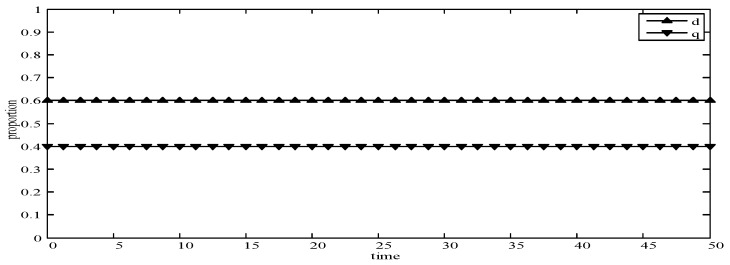
Evolution curves of *d* = 0.6 and *q* = 0.4.

**Figure 6 sensors-19-02927-f006:**
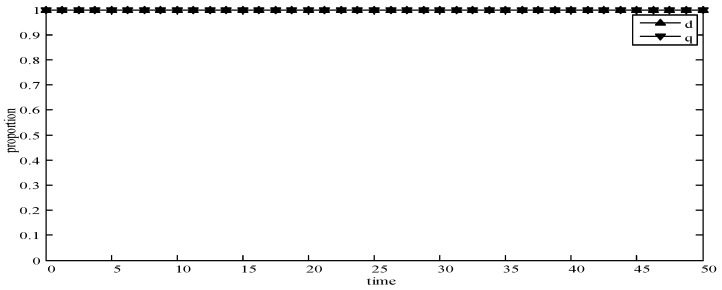
Evolution curves of *d* = 1 and *q* = 1.

**Figure 7 sensors-19-02927-f007:**
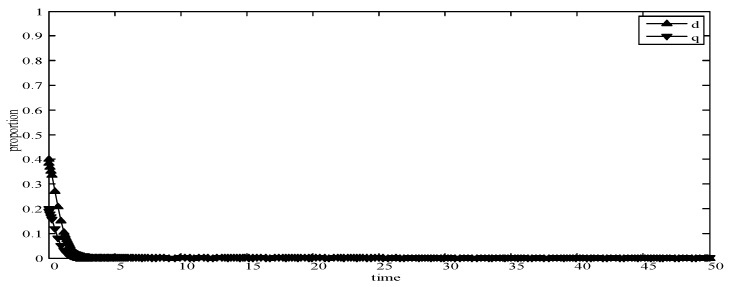
Evolution curves of *d* = 0.4 and *q* = 0.3.

**Figure 8 sensors-19-02927-f008:**
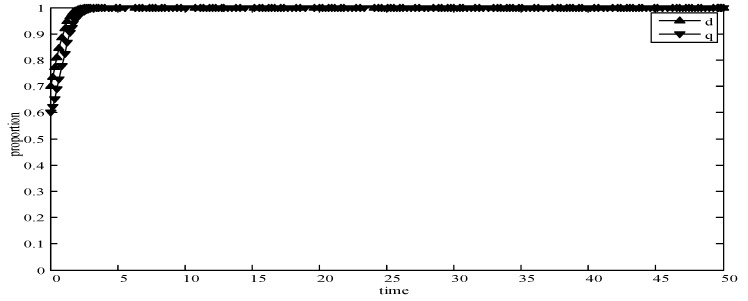
Evolution curves of *d* = 0.7 and *q* = 0.6.

**Figure 9 sensors-19-02927-f009:**
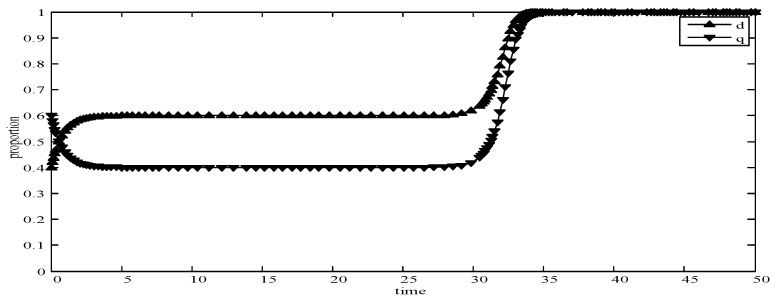
Evolution curves of *d* = 0.4 and *q* = 0.6.

**Figure 10 sensors-19-02927-f010:**
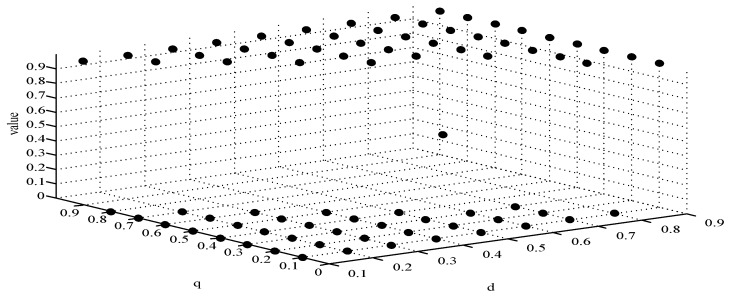
Selection of different initial state strategies.

**Table 1 sensors-19-02927-t001:** Values of the Jacobi determinant and trace.

Equilibrium Point	Determinant and Trace	Det	Trace
$F_{1}(0,0)$	$\begin{array}{l} \det J_{F_{1}}=(b_{hl}-b_{hh})(a_{lh}-a_{hh})\\ \mathrm{trace}J_{F_{1}}=(b_{hl}-b_{hh})+(a_{lh}-a_{hh}) \end{array}$	+	-
$F_{2}(0,1)$	$\begin{array}{l} \det J_{F_{2}}=-(b_{ll}-b_{lh})(a_{lh}-a_{hh})\\ \mathrm{trace}J_{F_{2}}=(b_{ll}-b_{lh})-(a_{lh}-a_{hh}) \end{array}$	+	+
$F_{3}(1,0)$	$\begin{array}{l} \det J_{F_{3}}=-(b_{hl}-b_{hh})(a_{ll}-a_{hl})\\ \mathrm{trace}J_{F_{3}}=-(b_{hl}-b_{hh})+(a_{ll}-a_{hl}) \end{array}$	+	+
$F_{5}(1,1)$	$\begin{array}{l} \det J_{F_{5}}=(b_{ll}-b_{lh})(a_{ll}-a_{hl})\\ \mathrm{trace}J_{F_{5}}=-(b_{ll}-b_{lh})-(a_{ll}-a_{hl}) \end{array}$	+	-

**Table 2 sensors-19-02927-t002:** Index quantification.

Index	Quantification Range	
1. Response time ($c_{1}$)/ min	[10,90]	
2. Distance ($c_{2}$)/ m	[0,5000]	
3. Data integrity ($c_{3}$)	STATIONARY	[0.7,0.9]
	WALKING	[0.5,0.7)
	RUNNING	[0.3,0.5)
4. Data reliability ($c_{4}$)	QUIET	[0.75,0.9]
	NORMAL	[0.6,0.75)
	ALERT	[0.45,0.6)
	NOISY	[0.3,0.45)

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
