# Peer review of "PUEGM: A Method of User Revenue Selection Based on a Publisher-User Evolutionary Game Model for Mobile Crowdsensing"

_sensors, 2019, doi:10.3390/s19132927_

Round 1
Reviewer 1 Report
The problem examined by the paper is very interesting and timely. However the presentation of the work is not done properly. There are several aspects that remain unclear in terms of the modeling assumptions. Certain technical issues are not properly presented. In this way it is difficult to appreciate the novelty of the paper and verify the correctness of the results.
The authors introduce the idea of complex tasks, that is tasks that require data from different sensors. This is an interesting concept, however the presentation of the model does not properly introduce this concept. The games are characterized by the so-called quality of the data of each user and the so-called selection strategy of the publishers. The terminology needs to be better motivated and better explained to include all the concepts introduced in the text.
Given the limitations on the presentation of model, the authors introduce an algorithm for assessing the data quality. A pseudcode is presented assumes that the user data is given as input, however the notion of "user data" is not clear. Does this imply the data collected from the users or the quality of the data? The algorithm uses a sub-function the removal low-quality data, however no information is provided on how this is done.
In a similar way, the second algorithm that identifies the optimal user strategy, mentions as input the "publisher-user game tree", however there is no notion of how this tree is constructed. The authors briefly discuss about the time complexity and spatial complexity of the algorithm presented. However, they completely miss providing evidence and arguments on the correctness of the algorithm. It is unclear why the resulting output contains the optimal user strategies.
The performance evaluation conducted is missing several elements in order to be properly reproduced. The dataset should be properly cited. The classification method used is not properly described.
Beyond the repeatability of the experimental results, once again there are certain aspects that are not properly justified. For example, four indexes are provided, however it is unclear what purpose they serve and how the authors ended up with these indexes.
The authors need to improve certain aspects of the paper:
1. It is not clear if the Publisher-User Evolutionary Game Model (PUEGM) is a novelty of the paper or if it is background.
2. The authors need to explain the reason that the players have limited rationality. Maybe a new subsection would help.
3. The authors need to explain why the participants are iterating over their strategy, why they are changing their strategy over time and how evolutionary models fit in this concept.
4. The characterization of the data quality of each user is not clear. The data quality assessment methods needs to be better explained. In particular the way the quantification of each user data in data quality assessment indexes is not clear.
"Eq. 2 represents the calculation method of the error value of user data when the data quality assessment indexes are benefit index." - this is not clear.
"epsilon is a positive infinite decimal." - this is also unclear
5. "Different tasks often need various data. For example, in the fastest route search traffic
monitoring of rescue vehicles, the required data pays more attention to accuracy and effectiveness." - this sentence is not clear.
6. In Algorithm 1, the pseudo-code "Judge c_j" is not clear, neither the pseudo-code "sort the user data".
7. "However, in the evolutionary game, the ESS solution usually means that when the user change strategy, the entire system can be restored to a stable state." - This sentence is very confusing.
8. "This means that when the player is in the ESS state, the user’s optimal revenue can be achieved." - This is not clear.
9. The value of Figure 3, 4, 5 is not clear.
Reviewer 2 Report
The paper was quite interesting considering the problem of incentives for the users to participate in crowdsensing projects, however the idea itself is not new. The problems I found were mainly with the user data authors wrote about. As I understand the good quality data for such project is hard to obtain - the authors define user data, but they do not give information about the sample they have, what kind of data they used etc. This cause problems with assessing the experiment results, and quality of user data assessment and defining why the algorithm results are optimal. Also in Algorithm 2 authors mention the publisher-user game tree which is not mentioned and explained anywhere else.
I think these should be explained in order to publish the paper.
Overall I think that the paper is interesting and could be useful for people dealing with crowdsensing projects, however above mentioned problems require additional explanation. Also English needs a little work.
Reviewer 3 Report
The paper presents a model, based on game theory, of the evolution of the interactions between users and publishers in an MCS. The model makes use of standard model theory recipes and I fell that as a good point.
My main concern is about the applicability of such models. For instance, I don't understand how the numerical experiment based on data set (fixed) may bring any information about the relevance of incentive mechanisms. Such validation would require to consider a real dynamical (online) MCS. It is as well difficult for me to understand how users may impact the data quality? Is it expected that users measuring wifi traces will do anything in order to improve the data quality? Moreover, the concept of data quality in this exemple is not really stated.
In summary, I feel that the paper framework is interesting and relevant, I feel that the paper is sound and does not look over sophisticated. However, I feel that the model is hardly applicable to a real situation. Works that present evaluation of real online process would bring more information.
Round 2
Reviewer 1 Report
The authors have addressed all the comments made in the previous round. There are no further comments.